# Evaluation of Corrosion Behavior and In Vitro of Strontium-Doped Calcium Phosphate Coating on Magnesium

**DOI:** 10.3390/ma14216625

**Published:** 2021-11-03

**Authors:** Jung-Eun Park, Yong-Seok Jang, Ji-Bong Choi, Tae-Sung Bae, Il-Song Park, Min-Ho Lee

**Affiliations:** 1Department of Dental Biomaterials and Institute of Biodegradable Material and Oral Bioscience, School of Dentistry, Jeonbuk National University, Jeonju-si 54896, Korea; pje312@naver.com (J.-E.P.); yjang@jbnu.ac.kr (Y.-S.J.); submissi@naver.com (J.-B.C.); bts@jbnu.ac.kr (T.-S.B.); 2Division of Advanced Materials Engineering, Research Center for Advanced Materials Development and Institute of Biodegradable Materials, Jeonbuk National University, Jeonju-si 54896, Korea

**Keywords:** magnesium, strontium, calcium, phosphate, biodegradable, biocompatibility

## Abstract

This study investigated the biocompatibility of strontium-doped calcium phosphate (Sr-CaP) coatings on pure magnesium (Mg) surfaces for bone applications. Sr-CaP coated specimens were obtained by chemical immersion method on biodegradable magnesium. In this study, Sr-CaP coated magnesium was obtained by immersing pure magnesium in a solution containing Sr-CaP at 80 °C for 3 h. The corrosion resistance and biocompatibility of magnesium according to the content of Sr-CaP coated on the magnesium surface were evaluated. As a result, the corrosion resistance of Sr-CaP coated magnesium was improved compared to pure magnesium. In addition, it was confirmed that the biocompatibility of the group containing Sr was increased. Thus, the Ca-SrP coating with a reduced degradation and improved biocompatibility could be used in Mg-based orthopedic implant applications.

## 1. Introduction

Numerous studies have reported on magnesium (Mg) and magnesium alloys as implant materials because of their superior mechanical properties and biodegradation characteristics. Because magnesium implants are degraded in vivo, there is no need to perform secondary surgery in order to remove an implant. Moreover, magnesium and its alloys have a similar density, elastic modulus, and strength to those of natural bones; thus, they are appropriate for use as implant materials. However, bionic magnesium alloys developed thus far show fast corrosion and rapidly decreasing mechanical properties, and hence, their clinical application as implants is limited [1,2].

To overcome the challenges associated with magnesium, various studies have been conducted on surface modified by methods to enhance the physical properties and corrosion resistance of materials. Research has been carried out to delay the bio-absorption rate and improve biocompatibility of magnesium implants through alloying, by adding various nontoxic elements or by post-treatments such as solid solution treatment, rolling, and pressing [3]. Examples of surface modifying methods include various treatments such as physical vapor deposition, electrodeposition, anodic oxidation, and chemical vapor deposition [4]. Among these, chemical vapor deposition is simple, economical, and an effective method of coating the surface of an implant with a complicated structure.

Chemical coating of a magnesium surface aims to protect its surface by using inorganic, organic, or hybrid coating [3]. In particular, bioceramic coating which shows superior biocompatibility can improve the corrosion resistance and biocompatibility of a metal implant. Bioactive calcium phosphate (CaP), the main component of bone, has been reported to enhance the corrosion resistance and biocompatibility of implants after being coated using methods such as chemical deposition, anodization, and electrodeposition [5].

Strontium (Sr) is a trace element found in natural bone, and it increases the differentiation of osteoblasts and impedes the growth and proliferation of osteoclasts [6,7]. Furthermore, strontium strengthens osteogenesis, reduces osteolysis, and controls bone remodeling [8,9]. Strontium cation can be included in compositions of bone-like materials such as hydroxyapatite (HAP), i.e., the main component of bone, by replacing the calcium cation, and the crystallinity, lattice energy, and solubility vary depending on its doping degree [10,11]. According to previous studies, Sr-substituted calcium phosphate shows osteoinduction activity and higher solubility compared with calcium phosphate [12]. Because a doping amount of strontium higher than the limiting concentration in calcium phosphate has a negative effect on proliferation, growth, and differentiation of osteoblasts, it is important to find the optimum doping amount [13,14,15].

Strontium has been introduced in metal coatings for applications in biomaterials, and various studies have reported the effect of doped Sr content in calcium phosphate coating on the biocompatibility of metallic materials [14,16,17]. According to previous studies, Sr-CaP coating utilizing micro-arc oxidation enhances the osseointegration of titanium and restrains the differentiation of osteoclasts [18]. Moreover, Sr-CaP coating on a magnesium surface improves its corrosion resistance and osteogenesis differentiation [16,17]. Although calcium phosphate coating on a magnesium surface has been widely studied, only a few studies have reported strontium-substituted CaP coating on a magnesium surface using the chemical immersion method. Accordingly, this study utilized the chemical immersion method to coat Sr-CaP on a pure magnesium surface. The effects of strontium content in the coating solution on the properties and corrosion resistance of the coated magnesium surface were investigated. Furthermore, through cytotoxicity testing, the effect of the Sr content on biocompatibility was studied.

## 2. Materials and Methods

### 2.1. Sample Preparation

The samples used in this study were prepared by cutting magnesium (Mg; pure magnesium) into 10 mm × 10 mm × 2 mm dimensions. All samples were polished with SiC sandpapers in the order of #600~#2000. After polishing, the samples were sonicated in ethanol and distilled water for 5 min each to remove any remaining impurities and then were dried.

### 2.2. Deposition Solution Preparation

Deposition solution to coat the magnesium surface was prepared by dissolving Ca(NO_3_)_2_·4H_2_O (Sigma-Aldrich, Saint Louis, MO, USA), NaH_2_PO_4_·2H_2_O (Sigma-Aldrich, Saint Louis, MO, USA), and Sr(NO_3_)_2_ (Sigma-Aldrich, Saint Louis, MO, USA) in distilled water. The exact concentration of each content is tabulated in Table 1.

### 2.3. Surface Treatment

Polished magnesium specimens were deposited in the aqueous solution described in 1.2.2 and were hydrothermally treated at 80 °C for 3 h. The specimens were then rinsed with distilled water and dried to obtain surface-treated magnesium. The surface-treated samples were designated as 0Sr, 0.5Sr, 1Sr, or 2Sr, depending on the amount of strontium content in the deposition solution.

### 2.4. Surface Properties

#### 2.4.1. Surface Characterization

To characterize the surface-treated Sr-CaP layer on magnesium, a field emission scanning electron microscope (FE-SEM; SU-70, HITACHI, Tokyo, Japan) was utilized.

X-ray diffractometer (XRD; X’PERT-PRO Powder, PANalytical, Eindhoven, Netherlands) was used to analyze the crystal phases of the magnesium coating layer by scanning from 20° to 60° at 2°/min.

Fourier-transform infrared spectroscopy (FT-IR; Frontier, Perkin Elmer, Waltham, MA, USA) was used to verify the presence of functional groups in the surface-treated magnesium. The scanning range was set at 1000 to 4000 cm^−1^.

The roughness of the coated surface on magnesium was measured using a light meter (SURFTEST SV-3000, Mitutoyo, Kawasaki, Japan). Each sample was measured 10 times, and the results were statistically analyzed with the ANOVA method (one way, * *p* > 0.05).

#### 2.4.2. Corrosion Resistance Evaluation

Electrochemical impedance spectroscopy (EIS) and potentiodynamic polarization were conducted using a Parstat 2273 potentiostat/galvanostat (Ametek, OakRidge, TN, USA). Hanks’ balanced salt solution (HBSS; H2387, Sigma-Aldrich, St. Louis, MO, USA) was utilized as an electrolyte, and the three-electrode method was used for the measurement. Magnesium, platinum, and Ag/AgCl electrode (Orion, Beverly, MA, USA) were used as the working electrode, counter electrode, and reference electrode, respectively. An electrochemical corrosion test was carried out at room temperature at a scan rate of 3 mV/s. Electrochemical impedance spectroscopy was performed from 0.1 Hz to 10^5^ Hz.

### 2.5. Biocompatibility Evaluation

#### 2.5.1. Immersion Test

Immersion test for pure magnesium and Sr-CaP coated magnesium were conducted following the ASTM G31-72 standard [19]. Each sample was immersed in simulated body fluid (SBF) and the composition of the surface was investigated. The SBF was produced by adding 0.285 g/L of calcium chloride dehydrate, 0.09767 g/L of magnesium sulfate, and 0.350 g/L of sodium hydrogen carbonate into Hanks’ solution, and the pH was adjusted to 7.4. Each sample was immersed in SBF for 14 days. The SBF was replaced every three days. The pH change of the SBF was measured for 3 days. The precipitation pattern of hydroxyapatite after immersion was analyzed by XRD. In addition, platinum coating followed by FE-SEM were conducted to identify the composition of surface-coating layer.

#### 2.5.2. Cytotoxicity Evaluation

In this study, the MC3T3-E1 cell line was obtained from American Type Culture Collection (ATTC, Manassas, VA, USA). The magnesium sample prepared according to ISO 10993.12 [20] was eluted in a cell culture media in a 37 °C incubator for 3 days. The culture media was prepared by adding fetal bovine serum (FBS; Gibco Co., New York, NY, USA) with a 10% including antibiotic (penicillin) to α-MEM (Gibco Co., New York, NY, USA) medium. The cells were cultured at 37 °C in the 5% CO_2_ incubator (3111, Thermo Electron Corporation, Waltham, MA, USA).

##### Water-Soluble Tetrazolium Salt (WST) Assay

For WST assay, MC3T3-E1 cells were cultured on a 48-well plate at a cell density of 1.5 × 10^4^ cells mL^−1^ for 1 and 3 days. After 1 and 3 days of culture, the medium was removed. A total of 400 μL of a mixture of Cell Counting Kit-8 (CCK-8; Enzo Life Science Inc., Farmingdale, NY, USA) reagent and the α-MEM medium was added on each well containing sample, and the samples were kept in a 5% CO_2_ incubator. After 1 h, 100 μL of the reaction solution were placed on the 96-well plate and the absorbance was measured at 450 nm using an ELISA reader (Molecular devices, EMax, San Jose, CA, USA).

##### Cell Morphology

Under the same conditions as the WST assay, the medium was removed after the cell culture and samples were rinsed with phosphate buffered saline solution (PBS). Fixative solution (3% formaldehyde + 0.2% glutaraldehyde) was added, maintained for 20 min, and removed. Next, fix the cells were rinsed with PBS. The cells were colored with a dye (0.3% crystal violet) and an optical microscope (DM2500, Leica, Wetzlar, Germany) was used to observe the morphology of the cells.

##### Alkaline Phosphatase (ALP) Activity

ALP activity was evaluated using a TRACP & ALP assay kit (TakaRa, Shiga, Japan). MC3T3-E1 cells were cultured on a 48-well plate with a cell density of 1.5 × 10^4^ cells mL^−1^ for 7 and 14 days. After culture, the medium was removed and the plate was rinsed with saline solution. The extraction and ALP buffer solutions adding p-Nitrophenyl Phosphate (pNPP) solution reacted with the cells in a 37 °C incubator for 1 h. 100 μL of reaction solution was placed on a 96-well plate, and the absorbance was measured at 405 nm using an ELISA reader.

#### 2.5.3. Cytokine Quantification

The cell type used in this study was a mouse macrophage cell line, Raw 264.7 (Cell Line Bank, Seoul, Korea). The culture medium was made by adding 10% fetal bovine serum (FBS; Gibco Co., New York, NY, USA) with containing antibiotic (penicillin) to D-MEM medium (Gibco Co., New York, NY, USA). The cell culture was conducted in an 5% CO_2_ incubator (3111, Thermo Electron Corporation, Waltham, MA, USA) at 37 °C. The culture was carried out in a 12-well plate at a concentration of 4.0 × 10^5^ cells well^−1^ for 6 h. Subsequently, the medium was removed and replaced with D-MEM (with 0.5% FBS) and the cell culture was performed for 18 h. In this study, tests were carried out by extracting the magnesium sample in the cell culture medium (D-MEM with 0.5% FBS) according to ISO 10993.12. Each extracted medium was replaced, 0.5 µg/mL of lipopolysaccharide (LPS) solution was added to each well, and the samples were cultured for 7 days. The control groups were D-MEM (with 0.5% FBS) and D-MEM (with 0.5% FBS) with 50 µg/mL of LPS added. For TNF-α and IL-1, the cultured media were collected and subjected to centrifugation (2500 rpm, 2 min). Using supernatant, TNF-α and IL-1 were measured with TNF-α and an IL-1 kit (BMS607-3 and BMS6002, Thermo Fisher, Cleveland, OH, USA). The absorbance of the supernatant was measured at 450 nm using a microplate spectrophotometer. The macrophage cell was stained with 0.3% crystal violet and its morphology was determined by optical microscopy.

### 2.6. Statistical Analysis

All experiments were performed five times. Statistical analysis was conducted by one-way analysis of variance with Tukey test (SPSS ver 21.0 software, IBM SPSS Statistics, Chicago, IL, USA). Data are presented as the mean ± SD. A *p* value of less than 0.05 was considered to be significant (* *p* < 0.05). When there is no significant difference, it was marked with ns (no statistical significance).

## 3. Results

To identify the crystal structure of the surface coated with XSr-CaP depending on the magnesium and strontium content, X-ray diffractometer (XRD) results are shown in Figure 1A. Only the magnesium peak was observed on the surface of magnesium. In the surface-treated group, a magnesium peak and a CaHPO_4_·2H_2_O (DCPD) peak were observed, but no peak related to Sr compounds was clearly observed. It was not clearly observed because it overlapped with the DCPD peak.

To study the functional groups in the coating layer on the magnesium surface, FT-IR analysis was conducted, the results of which are presented in Figure 1B. No peaks were found from the surface of the pure magnesium with no surface treatment. On the XSr-CaP coated magnesium surface with varying Sr content, peaks were found at 555 cm^−1^, 529 cm^−1^, 882 cm^−1^, 989 cm^−1^, 1059 cm^−1^, and 1125 cm^−1^. These peaks are attributed to absorption by phosphate (PO_4_^3−^). The peak at 1637 cm^−1^ corresponds with H_2_O. Furthermore, the peak at 1344 cm^−1^ corresponds to carbonate (CO_3_^2−^). Therefore, it was verified that XSr-CaP is coated on the magnesium surface. There was little difference in the peaks displayed at differing Sr content.

Figure 2A shows FE-SEM images of the surface and coating layer of the Sr-doped Ca-P coated magnesium. The surface of the magnesium is densely covered with a fish-scale-like structure (Figure 2A). The coated surface is homogeneous and appears to have no defects. Furthermore, as the Sr content increases, the surface has a finer structure. Cross-sectioned of each surface was to confirm the coating layer. The thickness of the coating layer was approximately 25.4 μm for 0Sr, approximately 21.8 μm for 0.5Sr, approximately 20 μm for 1Sr, and approximately 17.9 μm for 2Sr. As the content of strontium increased, the thickness of the coating layer became thinner. Moreover, the 1Sr group showed the most uniform coating layer.

Figure 2B shows the mapping results from the cross-sectional image of the 1Sr group by FE-SEM, which enables identification of the elementary composition of the coating layer. The coating layer appears uniform, and the Sr, Ca, and P elements are evenly distributed. This confirmed that the calcium phosphate doped with strontium was densely covered on the magnesium surface.

Figure 2C shows the results of surface roughness measurements of each sample. Compared to the surface roughness of the pure magnesium, the surface roughness of the Sr-CaP coated samples increases. Comparison within the Sr-CaP coated samples shows that increasing Sr content tends to decrease the surface roughness. As the Sr content increases, nucleation occurs rapidly and the crystals of Sr-CaP become dense, thus the thickness of the coating layer becomes thinner and the roughness of the surface decreases.

Figure 3A shows the potentiodynamic polarization test results of each group in Hanks’ solution. The corrosion potential (E_corr_) value for PM is −1.706 V, and those of the surface-treated 0Sr, 0.5Sr, 1Sr, and 2Sr are −1.546 V, −1.566 V, −1.559 V, and −1.549 V, respectively. The PM showed the highest value of corrosion current at 1.439 × 10^−4^ A/cm^2^, and those of the surface-treated 0Sr, 0.5Sr, 1Sr, and 2Sr were 8.725 × 10^−6^ A/cm^2^, 6.838 × 10^−6^ A/cm^2^, 7.949 × 10^−6^ A/cm^2^, and 5.208 × 10^−6^ A/cm^2^, respectively. Compared with pure magnesium, the corrosion potential significantly increases and the corrosion current density decreases due to the surface treatment, but there is no significant difference between the surface-treated groups with different strontium doping amounts.

To investigate the corrosion characteristics of the coating layer according to its structural and electrochemical properties, electrochemical impedance spectrometry (EIS) was carried out in Hanks’ solution. Figure 3B shows the modeled resistance of the Nyquist plot for pure magnesium and Sr-CaP surface-treated magnesium in Hanks’ solution. The corresponding equivalent circuits are shown in Figure 3B. The point where a semicircle is drawn in the Nyquist plot represents the corrosion resistance. All groups show a capacitive loop. The induction resistance loop was not observed. The polarization resistance was the highest in the 2Sr group.

Figure 3C shows the pH changes observed after immersion of pure magnesium and Sr-CaP coated magnesium in Hanks’ solution at 37 °C for 72 h. When the surface-treated magnesium groups were compared with pure magnesium, they showed a lower pH value. There was no significant difference in the pH values among the surface-treated groups, but the 2Sr group showed the lowest pH value.

Figure 4A is an image of the surfaces of pure magnesium and Sr-CaP coated magnesium observed by FE-SEM after immersed them in Hanks’ solution at 37 °C for 14 days. A precipitate with the shape of apatite was observed on the surface of all samples, and in particular, more precipitates were present on the surface of the Sr-CaP coated samples compared with that of the nontreated sample.

Figure 4B shows the XRD pattern from pure magnesium and Sr-CaP coated magnesium after immersed them in Hanks’ solution at 37 °C for 14 days. From the pure magnesium, only the magnesium peaks are observed. The magnesium peaks are weaker in the Sr-CaP coated magnesium, and a CaHPO_4_·2H_2_O and Ca_10_(PO_4_)_6_(OH)_2_ peak is observed. In addition, an increase in the Sr content results in a peak with a sharper shape.

The MC3T3-E1 cells were cultured for 1 and 3 days in the medium in which the pure magnesium and the Sr-CaP coated magnesium were extracted, and cell proliferation was measured (Figure 5A). On the basis of the results, more cell proliferation took place in the Sr-CaP coated samples than with the pure magnesium after 1 day, and there was no statistically significant difference found among the Sr-CaP coated samples. After 3 days of cell culture, the cell proliferation increased in all samples compared to the 1-day results, and more cell proliferation was found in the Sr-CaP coated group than in the pure magnesium. Furthermore, increasing the Sr content increased cell proliferation.

The MC3T3-E1 cells were cultured for 7 and 14 days in the medium in which the pure magnesium and the Sr-CaP coated magnesium were eluted, and the ALP activity of cells was investigated (Figure 5B). After 7 days, the ALP activity of the Sr-CaP coated magnesium group was lower than that of the pure magnesium, and the ALP activity decreased with increasing Sr content. After 14 days of cell culture, the ALP activity of the Sr-CaP coated magnesium was higher than that of the pure magnesium. Moreover, increasing the Sr content led to a lower ALP activity.

Figure 5C shows the cell morphologies after culturing the MC3T3-E1 cells in medium that eluted Sr-CaP coated magnesium with different Sr contents for 1 and 3 days. On day 1, cell shape showed a round shape with developed cytoplasm in all groups. On day 3, the cell grew densely in all groups due to many cell proliferations.

Figure 6 presents the formation of Il-1 (A) and TNF-α (B), which are proinflammatory cytokine genes. The concentration of TNF-α in the culture supernatant increased in the order LPS− < 1Sr < PM < 0.5Sr < 0Sr < 2Sr < LPS+. The concentration of IL-1 increased in the order PM < 0.5Sr < 2Sr < 1Sr < 0Sr < LPS− < LPS+.

Figure 6C shows the shape of the raw cells. LPS−, which does not contain an inflammation inducer, was found to have a round shape. In the LPS+ group, to which inflammation inducer was added, cell legs were developed, and the shape of the cells was elongated. The 0Sr, 0.5Sr, 1Sr, and 2Sr also showed cell legs and had an elongated shape.

## 4. Discussion

In this study, it was hypothesized that the Sr-CaP layer could control the corrosion rate of magnesium and enhance biocompatibility. A Sr-CaP coating layer was produced on the magnesium surface using a chemical immersion process, and the surface properties and biocompatibility towards bone of the coated magnesium were evaluated.

This study aimed to investigate changes on the Sr-CaP coated magnesium surface. Based on the FT-IR results, functional groups that were not found on the pure magnesium surface were observed on the Sr-CaP coated magnesium surface (Figure 1B). First, peaks were observed at 882 cm^−1^, 989 cm^−1^, 1059 cm^−1^, and 1125 cm^−1^, which correspond to the absorption peaks of phosphate (PO_4_^3−^). The spectra of the deposits agree with the formation of dicalcium phosphate dihydrate (DCPD), having bands and shoulders attributed to the HPO_4_ stretching mode at 882 cm^−1^, 989 cm^−1^, 1059 cm^−1^, and 1125 cm^−1^ [21]. The peak at 1637 cm^−1^ is the stretching and bending mode of H_2_O [2]. The main absorption bands in the spectra can be assigned to P-O of the PO_4_ group at 529 and 555 cm^−1^, revealing the presence of HAP [22]. The peak at 1344 cm^−1^ is relevant to carbonate (CO_3_^2−^) and suggests that phosphate ion was partially replaced by carbonate ion [2]. The presence of a CO_3_ band reveals that the formed Ca-P is carbonated Ca-P, which has the same composition as bone apatite. Moreover, on the basis of the XRD results, a CaHPO_4_·2H_2_O phase was observed on the Sr-CaP coated magnesium surface (Figure 1A). DCPD (CaHPO_4_·2H_2_O), which is thermodynamically unstable and dissolves relatively readily under physiological conditions, is a precursor to the formation of HAP [23]. The presence of the DCPD phase in the coated Sr-CaP on magnesium is advantageous for the condition of forming bones on the implant surface. The observation of the DCPD phase rather than the HAP phase proves that it has a higher solubility in physiological solution when more Sr is introduced into the HA structure [24,25]. In other words, the stability of the calcium phosphate varies with increasing Sr substitution. According to previous studies, an increase in Sr substitution in calcium phosphate reduces the particle size [16,26]. The results of this study are also in line with this observation, demonstrating a smaller particle size with increasing Sr content based on the FE-SEM results (Figure 2A). As the content of strontium increased, nucleation occurred faster, the shape of the surface became dense, and the surface roughness decreased. Due to the denser coating layer, the thickness of the coating layer was reduced.

The chemical composition and compactness of the coating play important roles in controlling the decomposition rate of the sample [27]. The Sr-CaP coating layer provides instant protection in the SBF, and thus, the sample decomposition reaction is slower than that in uncoated pure magnesium. If the coating is soaked for a long time, SBF can reach the surface through gaps and cracks. In general, it is known that the cathodic polarization curve is related to the formation of hydrogen due to the deoxygenation reaction of water, while the anodic polarization curve is related to the decomposition of magnesium and the corresponding formation of magnesium ion [28]. In the potentiodynamic polarization graph, the current density can be obtained from the intersecting point of the slopes of the cathodic and anodic curves of the Tafel line. When the curves are extrapolated, the corrosion rate can be estimated from the point of intersection [29]. The 2Sr group had a higher corrosion potential value compared with pure magnesium, and its current density was the lowest compared to the other samples (Figure 3A). Corrosion resistance increases with a decrease in current density. Therefore, it was confirmed that increasing Sr content enhances corrosion resistance. On the basis of the results, the corrosion resistance of the Sr-CaP coated magnesium was superior to that of pure magnesium. Moreover, an increase in the Sr content improved the corrosion resistance. The diameter of the capacitive semicircle of the measured Nyquist plot is closely linked with the corrosion rate [30].

When obtaining the Faraday impedance using the AC EIS method, the impedance value can be obtained from the frequency and low current; thus, the surface degradation is less, and more information can be obtained compared with the potential polarization test method [31]. The polarization resistance value was obtained as a semicircle in Figure 3B Nuquist plot. According to electrochemical theory, the inverse of charge transfer resistance is proportional to the rate of corrosion [32]. That is, as the size of the capacitive loop increases, the charge transfer resistance decreases and the corrosion rate decreases, making the electron transfer process difficult on the surface of the metal solution. The amount of interface reaction taking place is related to the diameter of the impedance semicircle. In this study, as the Sr content increased, the size of the capacitive resistance loop increased. In particular, the polarization resistance of 2Sr group was the highest. All capacitive factors of the equivalent circuit are replaced with constant phase elements (CPE) and are shown as unstable capacitors because of non-ideal dielectric behavior [33,34]. There are three main resistance factors used in the modeling. Rs is the solution resistance; Rc is the coating resistance, is a stomatal resistance related to infiltration of the solution into the coating, and is parallel to the CPEc; and Rt is the charge transfer resistance, a parameter effective for identifying the protection characteristics of the coating, and is parallel to CPEt [33,35]. Based on these factors, all groups can be fitted to R_s_(Q_c_R_c_) of equivalent circuit. In this study, the coating layer was thin, so coating resistance loop was not observed. However, it is considered that the polarization resistance value was high due to the formation of a dense coating layer as Sr was added. These results demonstrate that the corrosion resistance of magnesium is improved due to the Sr-CaP coating. It is considered that corrosion resistance is improved because the coating layer of magnesium coated with Sr-CaP is dense.

The bioactivity of the Sr-CaP coated magnesium implant was also evaluated in vitro. Based on the bioactivity of the surface of the material 14 days after using Hanks’ solution, sphere-shaped apatite particles were formed in the coating layer of the Sr-CaP coated magnesium, showing superior bioactivity (Figure 4A). This crystalline phase is considered to be CaHPO_4_2H_2_O and HA phases, similar to the XRD results (Figure 4B). Phosphate generally has a negative charge in Hanks’ solution, hence it absorbs calcium ion, thereby forming apatite. Finally, amorphous calcium phosphate is precipitated on the coating layer [36], and the following reactions take place between the substrate and solution [37]:
Ca^2+^ + HPO_4_^2−^ + 2H_2_O → CaHPO_4_·2H_2_O(1)
10Ca^2+^ + 6HPO_4_^2−^ + 8OH^−^ → Ca_10_(PO_4_)_6_(OH)_2_ + 6H_2_O(2)


This mechanism supports the results of the cell activity. MC3T3-E1 osteoblasts from the mouse cranial bone is widely used in research related to the cell activity of osteoblasts. It has similar metabolic functions such as cell proliferation, differentiation, and calcification during the formation of osteoblasts in vivo [38].

The effect of Sr-CaP on the growth of osteoblasts was evaluated. Cell proliferation increased in the Sr-CaP surface-treated magnesium group compared with the control group (Figure 5A). Moreover, an increase in Sr content enhanced proliferation of the osteoblasts (Figure 5A). According to Huang et al. [39], CaP coating combined with Sr and gelatin shows stronger cell viability compared to CaP coating alone, and it was proven to have a positive impact on MC3T3-E1 cells. Mg^2+^ and Ca^2+^ ions affect cell activity [40,41]; Mg^2+^ ion is known to help the growth of new bone tissues and shorten fracture recovery time [9]. Moreover, the addition of Sr expedites the combination of cells and proteins through other surface cell acceptors and leads to an active condition for improving cell growth [8,9]. Based on a study by Makkar et al. [2], the Sr-CaP coating was proven to enhance the proliferation adhesion of MC3T3-E1 cells and accelerate the appearance of osteogenesis markers. Based on the results, the Sr-CaP coating was shown to be nontoxic and showed superior biocompatibility. Furthermore, according to previous research [9], Mg^2+^ ion was proven to accelerate the differentiation of hMSC using osteoblastic phenotypes, while the presence of Sr^2+^ ion did not accelerate this differentiation. In this study, cell differentiation also tended to decrease with increasing Sr content based on ALP activity results (Figure 5B). Specifically, higher Sr content does not delay cell differentiation; rather, the corrosion resistance of the magnesium is enhanced due to Sr-CaP coating, leading to lower magnesium ion content; thus, cell differentiation is lower compared with that in pure magnesium.

Macrophages are involved in the prevention of host and incrementality retention against external stimuli and play a key role in biodefense by producing and secreting various cytokines when an inflammatory response is present [42,43]. LPS is a component of the cytoderm of Gram-negative bacteria, causes an inflammatory response at the local site and in the whole body, and increases the formation of various inflammatory factors through the activation of numerous intracellular signal circuits including NF-k by combining with TLR4 of the macrophage; hence, it is a research model widely used in inflammatory response and anti-inflammatory drugs [44,45]. Cytokine is a protein secreted in immune cells and controls the immune system, and TNF-α and IL-1 are typical inflammatory cytokines secreted in active macrophages. TNF-α and IL-1 are important media for septic shock and play a role in accelerating the activation of inflammatory reactions [45]. This study investigated the impact of magnesium surface treatment on the formation of TNF-α (pro-inflammatory) and increase of IL-1 (interleukin-1) with inflammation-causing factors in Raw 264.7 cells. After culturing cells in the magnesium extraction solution, the content of these factors in the culture medium was measured. On the basis of the results of this study, the magnesium and surface-treated magnesium showed a lower content of inflammatory response materials compared with the control group, LPS+ (Figure 6). Furthermore, IL-1 showed even lower content of inflammatory response materials compared to LPS−.

Finally, the Sr-CaP coating on the magnesium surface showed a uniform and dense surface without cracks, and this coating improved the corrosion resistance of magnesium. In addition, as a result of in vitro tests, Sr-CaP coating improved bioactivity and cell proliferation.

## 5. Conclusions

Sr-CaP was coated on the pure magnesium surface using a chemical immersion method. The corrosion resistance properties and biocompatibility of the Sr-CaP coating were evaluated according to the Sr content coated on the magnesium surface.

The surface of magnesium showed a finer crystal pattern as the Sr content increased. Due to this dense coating layer, Sr-CaP coated magnesium showed lower current density values and higher polarization resistance than pure magnesium. Therefore, the corrosion resistance of Sr-CaP coated magnesium was improved. In addition, Sr-CaP coated magnesium showed apatite-like precipitate after immersion in Hanks’ solution, and bone-like substances such as hydroxyapatite were deposited. As a result, it was confirmed that the surface of Sr-CaP coated magnesium has improved bioactivity.

As a result of culturing MC3T3-E1 cells in Sr-CaP coated magnesium eluate, it was confirmed that the proliferation and cell differentiation of osteoblasts were improved. Moreover, as a result of confirming the inflammatory factor, the content of the inflammatory factor was low. In conclusion, the surface treatment conditions of the 1Sr group are considered to be the most optimal conditions in terms of corrosion resistance and biocompatibility. Therefore, Sr-CaP coating on magnesium using a chemical immersion method improved the corrosion resistance and biocompatibility towards bone of magnesium. Based on this, the coating of Sr-CaP on magnesium is suitable for medical implant applications.

## Figures and Tables

**Figure 1 materials-14-06625-f001:**
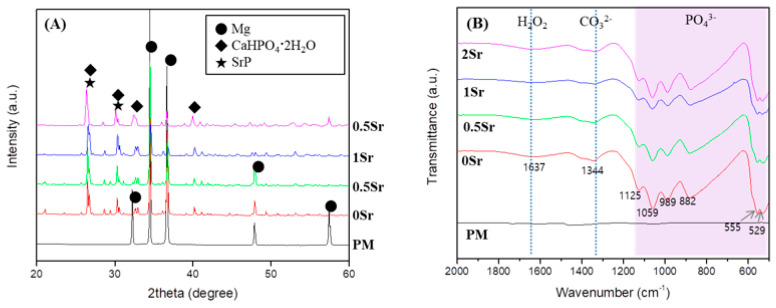
(**A**) XRD patterns; and (**B**) FT-IR spectrum of uncoated, Ca-P coated and Sr-doped Ca-P coated magnesium.

**Figure 2 materials-14-06625-f002:**
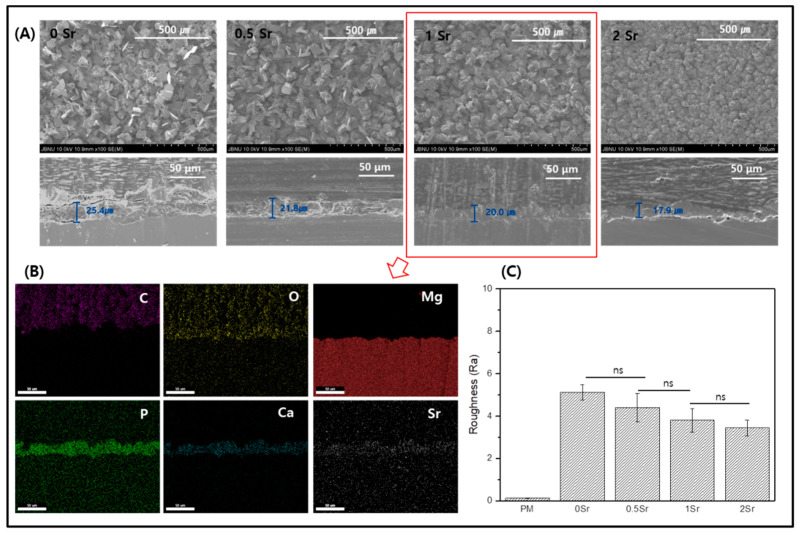
(**A**) Field emission scanning electron microscope (FE-SEM) image of surface and cross section, (**B**) Mapping image of cross-section of 1Sr coated Mg, and (**C**) Roughness of surface (ns, no statistical significance).

**Figure 3 materials-14-06625-f003:**
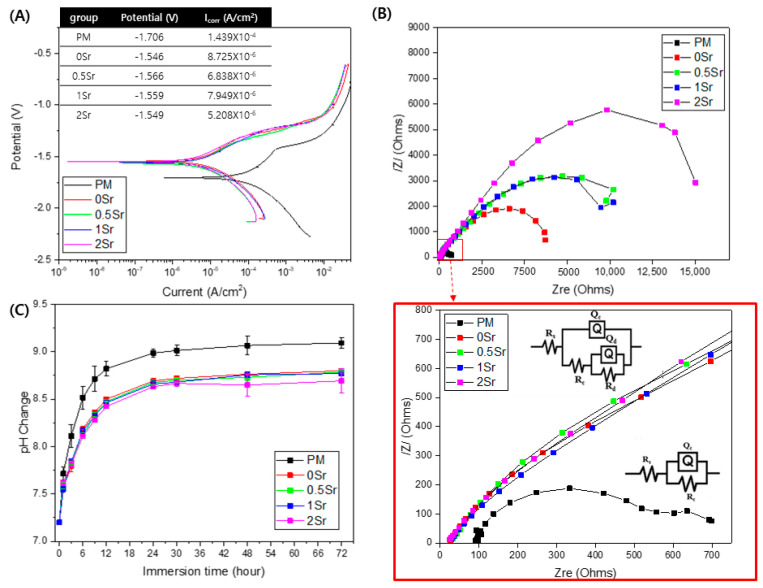
(**A**) potentiodynamic polarization curve obtained in Hanks’ solution, (**B**) Nyquist plots from EIS curves of surface, and (**C**) pH change of untreated and treated magnesium groups of according to immersion in Hanks’ solution for 72 h.

**Figure 4 materials-14-06625-f004:**
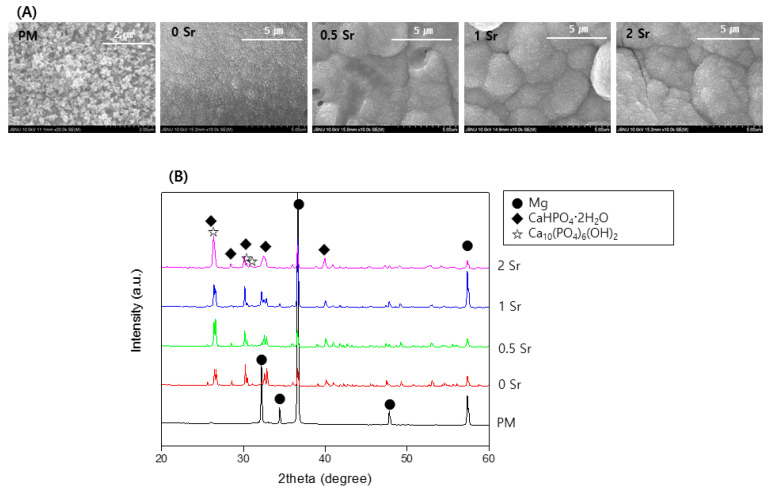
(**A**) FE-SEM images of the surface and (**B**) XRD patterns after immersion in Hanks’ solution for 14 days.

**Figure 5 materials-14-06625-f005:**
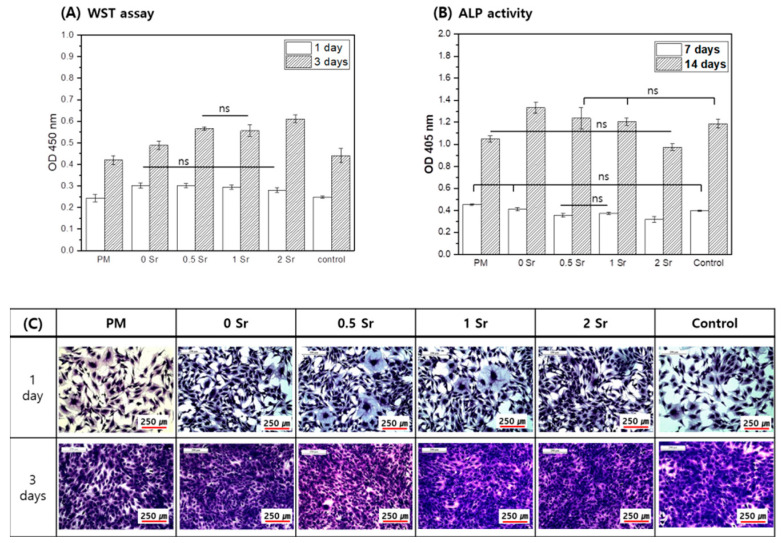
Proliferation (**A**), Differentiation (**B**), and Morphology (**C**) for MC3T3-E1 cells cultured in the extracted media; analyzed by WST assay and crystal violet stanin after 1 and 3 days of culture (**A**,**C**) and by ALP activity after 7 and 14 days of culture (**B**), (ns, no statistical significance).

**Figure 6 materials-14-06625-f006:**
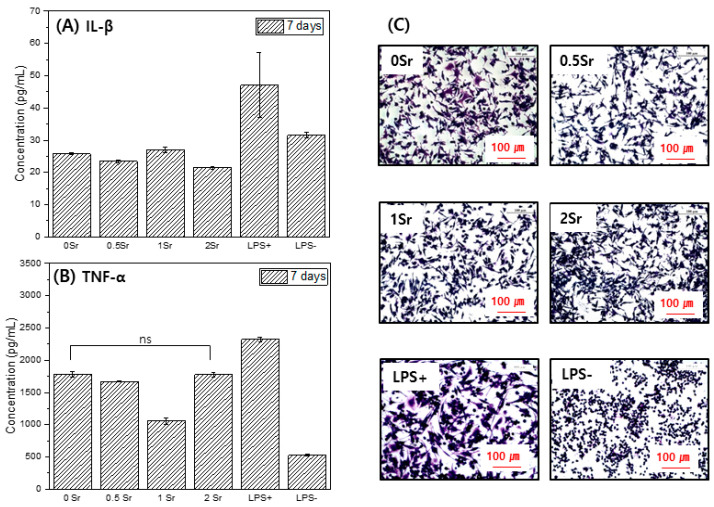
IL-1(**A**), TNF-α(**B**), and cell morphology (**C**) for Raw 264.7 cells cultured, in the extracted media for 7 days (ns, no statistical significance).

**Table 1 materials-14-06625-t001:** Detailed conditions for surface treatment.

Group	Ca(NO_3_)_2_·4H_2_O (g/L)	NaH_2_PO_4_·2H_2_O (g/L)	Sr(NO_3_)_2_ (g/L)	Temperature	Time
0Sr	39.44	15.6	-	80 °C	3 h
0.5Sr	37.55		1.69		
1Sr	34.71		4.23		
2Sr	31.64		6.98		

## Data Availability

Not applicable.

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
