# Peer review of "Evaluation of Corrosion Behavior and In Vitro of Strontium-Doped Calcium Phosphate Coating on Magnesium"

_materials, 2021, doi:10.3390/ma14216625_

Round 1
Reviewer 1 Report
This is a worthwhile paper that reports useful results on the effect of a strontium-doped calcium phosphate coating of magnesium in terms of its corrosion resistance and potential bone-contact biocompatibility. It its the basis of a sound publication but there are a few changes needed before the manuscript can be accepted. These are as follows:
Line 28: "was" should be "are" (i.e. change to both plural form and present tense).
Line 293: Biocompatibility is not a single property, but depends (among other things) on location in the body. Therefore this should be changed to "... biocompatibility towards bone..."
Lines 361 and 365 (and elsewhere in the text). There is inconsistency in the use of Hank's and Hanks' for the solution. Since it is named after the microbiologist John Hanks, it should be written Hanks'. Please check this throughout the text and correct where necessary.
Lines 378 and 384: Please omit the authors' given names, i.e. quote them simply as Huang and Makkar respectively.
Line 429. The comment about biocompatibility applies here as well. This should refer to "bone-contact biocompatibility...".
Subject to these corrections, the paper can proceed to publication.
Author Response
- Response to the comments of the reviewers
The comments of the reviewers were much useful to revise our manuscript. As suggested by the reviewers, all the necessary deletions/additions/corrections/modifications were carried out in the revised manuscript (please refer the portions highlighted in yellow color).
Comment: Line 28: "was" should be "are" (i.e. change to both plural form and present tense).
Response: We agree with the reviewer’s remark. As suggested by the reviewer, we changed from “was” to “are” in the revised manuscript.
Comment: Line 293: Biocompatibility is not a single property, but depends (among other things) on location in the body. Therefore this should be changed to "... biocompatibility towards bone..."
Response: We agree with the reviewer’s remark. As suggested by the reviewer, we changed from “biocompatibility” to “biocompatibility towards bone” in revised manuscript.
Comment: Lines 361 and 365 (and elsewhere in the text). There is inconsistency in the use of Hank's and Hanks' for the solution. Since it is named after the microbiologist John Hanks, it should be written Hanks'. Please check this throughout the text and correct where necessary.
Response: We agree with the reviewer’s remark. As suggested by the reviewer, we changed from “Hank`s” to “Hanks`” in the revised manuscript.
Comment: Lines 378 and 384: Please omit the authors' given names, i.e. quote them simply as Huang and Makkar respectively.
Response: We agree with the reviewer’s remark. As suggested by the reviewer, we changed to simply as Huang and Makkar in the revised manuscript.
Comment: Line 429. The comment about biocompatibility applies here as well. This should refer to "bone-contact biocompatibility...".
Response: We agree with the reviewer’s remark. As suggested by the reviewer, we changed from “biocompatibility” to “biocompatibility towards bone” in revised manuscript.

Reviewer 2 Report
The article presents the evaluation of corrosion behavior and biological performance in vitro of strontium-doped calcium phosphate coating deposited by chemical immersion method on pure magnesium. The introduction, experimental materials and procedures are well described. However, all the figures are too small to be clear, and they should be increased. A few comments below.
- There is no clear evidence of the presence of the SrP phase in the Sr-doped coatings. As can be seen from Fig. 1 (a), the SrP phase corresponds to the same peak as the DCPD phase.
- What is the amount of strontium contained in the corresponding samples 0.5Sr, 1Sr and 2Sr?
- There is no experimental evidence, e.g. EDX mapping or XRD, of this sentence (line 199-200): “The surface of the magnesium is densely covered with calcium phosphate doped with crystalline strontium”.
- Why does an increase in the Sr content in the solution lead to morphological transformations on the coating surface, a decrease in the thickness and roughness of the coating?
- The authors incorrectly believe only by one very weak peak in Fig. 4b that hydroxyapatite is present in the coatings after the immersion test. At least three reflexes of the detected phase should be present on the XRD patterns.
- There is not enough discussion about the relationship between the coatings structure, morphology, phase composition, elemental composition and their corrosion resistance and biological reaction in vitro. The authors limited the discussion only from the perspective of Sr concentration.
- In conclusion, it is unclear which concentration of Sr in solution is better?
Author Response
- Response to the comments of the reviewers
The comments of the reviewers were much useful to revise our manuscript. As suggested by the reviewers, all the necessary deletions/additions/corrections/modifications were carried out in the revised manuscript (please refer the portions highlighted in yellow color).
Comment: There is no clear evidence of the presence of the SrP phase in the Sr-doped coatings. As can be seen from Fig. 1 (a), the SrP phase corresponds to the same peak as the DCPD phase.
Response: We agree with the reviewer’s remark. As suggested by the reviewer, the surface-treated group, a magnesium peak and a CaHPO4·2H2O (DCPD) peak were oberved, but no peak related to Sr compounds was clearly observed. It was not clearly observed because it overlapped with the DCPD peak. The result section is modified with the inclusion of these points in the revised manuscript.
Comment: What is the amount of strontium contained in the corresponding samples 0.5Sr, 1Sr and 2Sr?
Response: We agree with the reviewer’s remark. Table 1 shows the Ca, P and Sr contents contained in the coating solution for 0.5Sr, 1Sr, and 2Sr groups.
Comment: There is no experimental evidence, e.g. EDX mapping or XRD, of this sentence (line 199-200): “The surface of the magnesium is densely covered with calcium phosphate doped with crystalline strontium”.
Response: We agree with the reviewer’s remark. As suggested by the reviewer, “The surface of the magnesium is densely covered with calcium phosphate doped with crystalline strontium” sentence has been removed and the “The surface of the magnesium is densely covered with a fish scale-like structure” sentence has been corrected. Also added the “This confirmed that the calcium phosphate doped with strontium was densely covered on the magnesium surface” sentence (line:217)
Comment: Why does an increase in the Sr content in the solution lead to morphological transformations on the coating surface, a decrease in the thickness and roughness of the coating?
Response: We agree with the reviewer’s remark. As suggested by the reviewer, As the Sr content increases, nucleation occurs rapidly and the crystals of Sr-CaP become dense, thus the thickness of the coating layer becomes thinner and the roughness of the surface decreases. The conclusion section is modified with the inclusion of these points in the revised manuscript.
Comment: The authors incorrectly believe only by one very weak peak in Fig. 4b that hydroxyapatite is present in the coatings after the immersion test. At least three reflexes of the detected phase should be present on the XRD patterns.
Response: We agree with the reviewer’s remark. As suggested by the reviewer, the figure 4(b) was modified by adding marks in the revised manuscript.
Comment: There is not enough discussion about the relationship between the coatings structure, morphology, phase composition, elemental composition and their corrosion resistance and biological reaction in vitro. The authors limited the discussion only from the perspective of Sr concentration.
Response: We agree with the reviewer’s remark. As suggested by the reviewer, the discussion section is modified in the revised manuscript as follows. added the sentences “As the content of strontium increased, nucleation occurred faster, the shape of the surface became dense, and the surface roughness decreased. Due to the denser coating layer, the thickness of the coating layer was reduced. (line:331-333)”, “It is considered that corrosion resistance is improved because the coating layer of magnesium coated with Sr-CaP is dense. (line:375-376)”and”Finally, the Sr-CaP coating on the magnesium surface showed a uniform and dense surface without cracks, and this coating improved the corrosion resistance of magnesium. In addition, as a result of in vitro tests, Sr-CaP coating improved bioactivity and cell proliferation. (line:431-434)”
Comment: In conclusion, it is unclear which concentration of Sr in solution is better?
Response: We agree with the reviewer’s remark. As suggested by the reviewer, the surface treatment conditions of the 1Sr group are considered to be the most optimal conditions in terms of corrosion resistance and biocompatibility. The conclusion section is modified with the inclusion of these points in the revised manuscript.
